**Enhanced Terrestrial Runoff during Oceanic Anoxic Event 2 on the North Carolina Coastal Plain, USA**

Christopher M. Lowery[1], Jean M. Self-Trail[2], Craig D. Barrie[3]

[1]University of Texas Institute for Geophysics, Austin, TX, USA
[2]United States Geological Survey, Reston, VA, USA
[3]GeoMark Research, LTD, Houston, TX, USA

*Correspondence to*: Chris Lowery cmlowery@utexas.edu

**Abstract**

A global increase in the strength of the hydrologic cycle drove an increase in the flux of terrigenous sediments into the ocean during the Cenomanian–Turonian Oceanic Anoxic Event 2 (OAE2) and was an important mechanism driving nutrient enrichment and thus organic carbon burial. This global change is primarily known from isotopic records, but global average data don't tell us anything about changes at any particular location. Reconstructions of local terrigenous flux can help us understand the role of regional shifts in precipitation in driving these global trends. The proto-North Atlantic basin was one of the epicenters of enhanced organic carbon burial during OAE2, and so constraining terrigenous flux is particularly important in this region; however, few local records exist. Here, we present two new OAE2 records from the Atlantic Coastal Plain of North Carolina, USA, recognized with calcareous nannoplankton biostratigraphy and organic carbon isotopes. We use carbon/nitrogen ratios to constrain the relative contribution of marine and terrestrial organic matter; in both cores we find elevated contribution from vascular plants beginning just before OAE2 and continuing through the event, indicating a locally strengthened hydrologic cycle. Terrigenous flux decreased during the brief change in carbon isotope values known as the Plenus carbon isotope excursion, and then increase and remain elevated through the latter part of OAE2. TOC values reveal relatively low organic carbon burial in the inner shelf, in contrast to black shales known from the open ocean. Organic carbon content on the shelf appears to increase in the offshore direction, highlighting the need for cores from the middle and outer shelf.

## 1 Introduction

The Cretaceous was characterized by intermittent periods of enhanced organic carbon burial linked to widespread black shale deposition and anoxia, termed Oceanic Anoxic Events (OAEs; Schlanger and Jenkyns, 1976; Jenkyns 2010). Although OAEs were originally defined by the widespread occurrence of black shales (Schlanger and Jenkyns, 1976) they were soon found to be associated with positive carbon isotope excursions driven by the excess global burial of organic carbon and representing a perturbation of the global carbon cycle (Scholle and Arthur, 1980; Arthur et al., 1987; Jenkyns, 2010; Owens et al., 2017). OAEs eventually became linked with the emplacement of large igneous provinces (Tarduno et al., 1991; Whitechurch et al., 1992; Leckie et al., 2002; Snow et al., 2005; Turgeon and Creaser, 2008; Monteiro et al., 2012; McAnena et al., 2013), suggesting a causal mechanism for enhanced organic carbon burial. In the case of the Cenomanian–Turonian OAE2 (~ 94 Ma), the emplacement of the Caribbean Large Igneous Province (e.g., Snow et al.,2005) is associated with significant warming (e.g., Friedrich et al., 2012) and resulted in a strengthening of the hydrological cycle and an increase in the flux of nutrients to the oceans (Blättler et al., 2011; Pogge von Strandmann et al., 2013).

Carbon isotopes reveal global changes in organic carbon burial rates but don't tell us anything about where that organic matter was buried. This is important because local organic matter enrichment can vary significantly in both timing (e.g., Tsikos et al., 2004) and magnitude (e.g., Owens et al., 2018) during an OAE. Similarly, the calcium isotope proxy used by Blättler et al. (2011) and the lithium isotope proxy used by Pogge von Strandmann et al. (2013) to determine changes in global terrigenous flux to the oceans don't tell us anything about local patterns of terrigenous runoff. Presumably, like organic carbon burial, the hydrologic cycle did not increase uniformly, but instead some regions experienced a greater change than others. Unfortunately, few local records of changes in the hydrologic cycle during OAE2 have been documented. Van Helmond et al. (2014) used palynological and biomarker data from the Bass River core (Ocean Drilling Program Site 174X) on the coastal plain of New Jersey, USA, to document local warming associated with enhanced contribution of terrestrial organic matter during OAE2. While

this result clearly indicates a stronger hydrologic cycle during OAE2, it only represents a single locality.
Similar work from Wunstorf, Germany, in the Lower Saxony Basin, reveals a clear association between
terrigenous flux (measured by palynology and biomarker data) and black shale development, but this
association isn't limited to OAE2, with additional intervals of elevated terrigenous input and black shale
deposition continuing after the end of the carbon isotope excursion (van Helmond et al., 2015). In the
Western Interior Seaway of North America, increases in kaolinite (a clay mineral formed in humid
environments) during OAE2 may be the result of wetter conditions, but these trends may also be caused
by shifting sediment source areas (Leckie et al., 1998). Overall, these existing records paint an incomplete
picture.
To fully understand these trends, it is essential to develop similar datasets from additional
localities. Such work will allow a more geographically complete understanding of changes in
precipitation during OAE2 and thus provide a window into the mechanisms which drove hydroclimate
during the hottest part of the Cretaceous greenhouse. Here, we present two new OAE2 sections from
cores drilled by the United States Geological Survey (USGS) on the coastal plain of North Carolina, on
the Atlantic margin of North America (Figure 1). We use organic carbon isotopes and calcareous
nannoplankton biostratigraphy to identify the OAE2 interval and organic carbon/nitrogen (C/N) molar
ratios to detect changes in terrigenous flux. These cores are only the second and third OAE2 intervals
described on the Atlantic Coastal Plain after the Bass River core (Bowman and Bralower, 2005; van
Helmond et al., 2014) and thus also provide important context for the response of the inner shelf to
OAE2, filling in an important gap in an important region (e.g., Owens et al., 2018) during this well-
studied time interval.

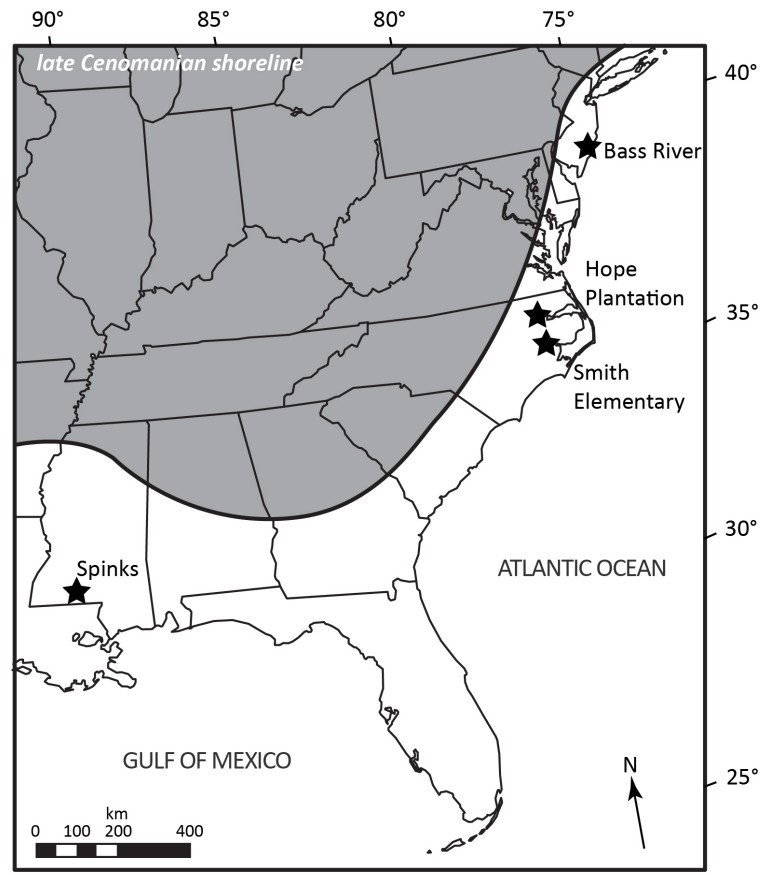


**Figure 1.** *Map of southeastern North America showing approximate late Cenomanian shoreline (land = grey) and the location of the cores discussed in this study. Shoreline position after Slattery et al. (2015) and Snedden et al. (2015).*

## 2 Geologic Setting

Cenomanian and Turonian sediments of the Atlantic Coastal Plain of the United States (Figure 1) are part of a sequence of strata that accumulated since the rifting of the Atlantic began in the Early Jurassic. However, study of the marine units in these sediments is difficult due to the absence of outcrops of this age and environment and their moderate to large burial depths in the Carolinas and Georgia (Sohl and Owens, 1991). Thus, their study is restricted to the limited number of available cores and/or cuttings, and their regional interpretations are often based on geophysical data obtained from water wells and scattered oil and gas test wells.

To the south, initial subsurface work in Florida and Georgia followed the nomenclature of the
Gulf Coastal Plain. Sediments in Georgia were variously attributed to the Cenomanian Woodbine
Formation, the Cenomanian/Turonian Eagle Ford Formation, and the Cenomanian/Turonian Tuscaloosa
Formation (Applin and Applin, 1944; Richards, 1945).  Applin and Applin (1947) later introduced the
name Atkinson Formation, with three unnamed members (upper, middle, and lower) for certain marine
and non-marine sediments in the subsurface of southern Alabama, southern Georgia, and northern
Florida. They correlated the lower member of the Atkinson to basal nonmarine sands and shales of the
coastal plain of Georgia, which they considered to be Cenomanian in age, and the middle member of the
Atkinson to the Tuscaloosa Marine Shale, which they considered to be Cenomanian/Turonian in age
(Applin and Applin, 1967).
Early work in South Carolina by Cooke (1936), Dorf (1952), and Heron (1958) considered
outcrops of the Middendorf Formation to be Cenomanian in age, based largely on stratigraphic position
and on long-ranging pollen and/or mollusks. Similarly, outcrops of the largely non-marine Cape Fear
Formation in North Carolina were attributed to the Cenomanian (Stephenson, 1912; Cooke, 1936).
Outcrops thought to be Turonian in age from both states were largely assigned to the Black Creek
Formation.
A shift in thinking regarding stratigraphic nomenclature was spurred by examination of sediments
from the Clubhouse Crossroads #1 core by Hazel et al. (1977), who found clear evidence of true
Cenomanian/Turonian marine sediments in the later-defined Clubhouse Formation near the base of the
downdip Coastal Plain section. Calcareous nannofossils and foraminifera of Cenomanian and Turonian
age were identified in the Clubhouse Formation (Hazel et al., 1977; Hattner and Wise, 1980; Valentine,
1984) and correlated with cuttings containing calcareous nannofossils from the Fripp Island, SC
deepwater well (Valentine, 1984). In North Carolina, Zarra (1989) reinterpreted the work of Spangler
(1950) using both foraminifera and sequence stratigraphic concepts, positively identifying Cenomanian
and Turonian sediments from the Esso #1 core and from cuttings of the Mobile #1, Mobile #2, Mobile #3,
and Marshall Collins #1 test wells. He used sediment and well log analysis to identify marginal marine
and inner shelf facies in the lower/middle Cenomanian and middle Turonian section, with a highstand in
the upper Cenomanian. These cores all contained a diverse assemblage of planktic foraminifera, including
species belonging to *Rotalipora, Praeglobotruncana, Dicarinella, Whiteinella,* and *Guembelitria* (Zarra,

1989).

This reevaluation ultimately resulted in the formal designation of the Cenomanian/Turonian
Clubhouse Formation (Gohn, 1992) in the Clubhouse Crossroads core. At the type locality, the Clubhouse
Formation consists of gray to gray-green, fine- to medium- grained, micaceous, muddy sands with flaser
to lenticular bedding and common bioturbation. Sequence stratigraphic analysis suggests that deposition
occurred in a shelf environment proximal to the shoreline and that these sediments represent latest
Cenomanian/earliest Turonian sea level rise prior to the early Turonian highstand event (Aleman
Gonzalez et al., 2020). The subsurface extent of this formation has now been documented across much of
South Carolina and North Carolina (Weems et al. 2007; Weems et al., 2019; Aleman Gonzalez et al.,

2020).

To the north, published documentation of marine Cenomanian/Turonian sediments from the mid-
Atlantic region appears to be limited to the E.G. Taylor No. 1-G well on the eastern shore of Virginia.
Valentine (1984) reports the presence of *Rotalipora greenhornensis,* which went extinct in the latest
Cenomanian, from one sample at 1520 ft.
Cenomanian/Turonian sediments of the northeast Atlantic Coastal Plain consist of the subsurface
Bass River Formation and its correlative updip equivalent, the Raritan Formation in Maryland, New
Jersey, and Delaware. The Bass River Formation is herein considered to be correlative with the
Clubhouse Formation of the southeastern Atlantic Coastal Plain. The Bass River Formation was first
described by Petters (1976) from the TC16 well in Bass River Township, New Jersey. It is considered to
be the fully marine equivalent of the Raritan Formation and is differentiated by its common shell material
and deeper water depositional environment (Miller et al., 1998). The Bass River Formation has variously
been assigned a late Cenomanian to early Turonian age in a variety of cores and wells based on
foraminifera (Petters, 1976, 1977; Miller et al., 1998; Sikora and Olsson, 1991), calcareous nannofossils
(Valentine, 1984; Miller et al., 1998; Self-Trail and Bybell, 1995), and ostracodes (Gohn, 1995). Miller
et al., (2004) document that the Bass River Formation was deposited predominantly in inner neritic to
middle neritic paleodepths.

**3 Methods**

**3.1 Study Sites**

The Hope Plantation core (BE-110-2004) was drilled by the USGS in April to May, 2004 in

Bertie County, North Carolina, on the property of Hope Plantation (36.0323°N; 78.0192°W) (Figure 1).
The hole was drilled as a stratigraphic test for Atlantic Coastal Plain aquifers, and was continuously cored
to a total depth of 333.6 m (1094.5 ft) below land surface. A suite of wireline logs, including natural
gamma ray and resistivity logs, were collected at the completion of drilling. Preliminary biostratigraphy
placed the marine Cenomanian/Turonian boundary interval between approximately 182.8-228.6 m (600-
750 ft). A summary of the general stratigraphy, downhole logging, and core images can be found in
Weems et al. (2007).

The Smith Elementary School core (CR-675) was drilled by the USGS in February and March,

2006 in Craven County, NC, on the grounds of the nominate school (35.2511°N; 77.2903°W) (Figure 1).
This hole was also drilled as a stratigraphic test for coastal plain aquifers, and was continuously cored to a
total depth of 323.1 m (1094.5 ft). Difficulties with the wireline tools and borehole stability limited the
collection of geophysical logs, and only a partial natural gamma ray log exists for the Clubhouse
Formation in this corehole. There, the marine interval that spans the Cenomanian/Turonian boundary is
between 288.3 and 323.1 m depth (945.9-1060.0 ft). Both cores are stored at the North Carolina
Geological Survey Coastal Plain core storage facility in Raleigh, NC, where we sampled them in May,

2019.

**3.2 Calcareous Nannofossils**

One hundred and ten samples from Hope Plantation and 84 samples from Smith Elementary School were examined for calcareous nannofossil content. Samples were taken from the central portion of broken core in order to avoid contamination from drilling fluid. Smear slides were prepared using the standard techniques of Bown and Young (1998) in samples with low total organic carbon (TOC); samples with increased TOC were prepared using the techniques of Shamrock et al. (2015) and Shamrock and Self-Trail (2016). Coverslips were affixed using Norland Optical Adhesive 61. Calcareous nannofossils were examined using a Zeiss Axioplan 2 transmitted light microscope at 1250x magnification under crossed polarized light. Light microscope images were taken using a Powershot G4 camera with a Zeiss phototube adaptor. Specimens were identified to the species level and correlated to the zonation schemes of Sissinghi (1977) and Burnett (1998), as modified by Corbett et al. (2014) for shelf settings.

**3.3 Foraminifera**

Ninety samples were prepared for examination of planktic and benthic foraminifera. Approximately 15 grams of material were soaked in a mixture of peroxide and borax for at least 24 hours, washed over a 63 μm sieve, dried overnight in an oven, and then examined for microfossils using a Zeiss Discovery V8 light microscope.

**3.4 TOC, C/N, and $\delta^{13}$C**

Core samples were analyzed for both their elemental composition (%C and %N) and organic carbon isotope signature ($\delta^{13}$C VPDB). To remove inorganic carbon content all of the material to be analyzed was initially washed with 1M hydrochloric (HCl) acid. There was no anticipated inorganic nitrogen content in the samples. All of the samples were analyzed on an elementar vario ISOTOPE select cube elemental analyzer (EA) connected to a VisION isotope ratio mass spectrometer (IRMS). The EA system follows dumas combustion and both generates and separates the gasses used for elemental composition determination and then releases the gas to the IRMS for isotopic determination. Every fifth

sample was run in duplicate and a check standard was run in triplicate every twentieth sample to ensure
the accuracy of the results. The elemental results were calibrated against a known sulfanilamide standard
and the precision of the results is +/-0.1% or better, and variation of duplicate samples was within range
of this uncertainty. The carbon isotope results were calibrated against four known reference standards
which cover the range of isotopic signatures expected in organic material (-15‰ to -35‰), and duplicates
and check standards were run at the same interval as above. All of the isotopic results are reported in per
mil (‰) relative to VPDB and the precision of the results is +/-0.1‰ or better.
**4 Results**
**4.1 Lithology**

Qualitative core descriptions are summarized below and in Figures 2 and 3. Broad

paleoenvironmental interpretations are based on lithology, paleontology, and stratigraphic relationships.
Benthic foraminifera, which are powerful tools to determine paleoenvironment in marginal marine
settings (e.g., Tibert and Leckie, 2004), are unfortunately absent here due to poor preservation (see
section 4.2 below). In both cores we recognize two informal members of the Clubhouse Formation: a
marine lower member characterized by bivalves, calcareous nannoplankton, finer grained sediments,
thinner beds, and sedimentary features common to inner neritic environments; and a marginal marine
upper member characterized by coarser grainsize, thicker beds, and woody plant debris instead of
calcareous marine fossils, indicating deposition in a delta front or distributary environment.

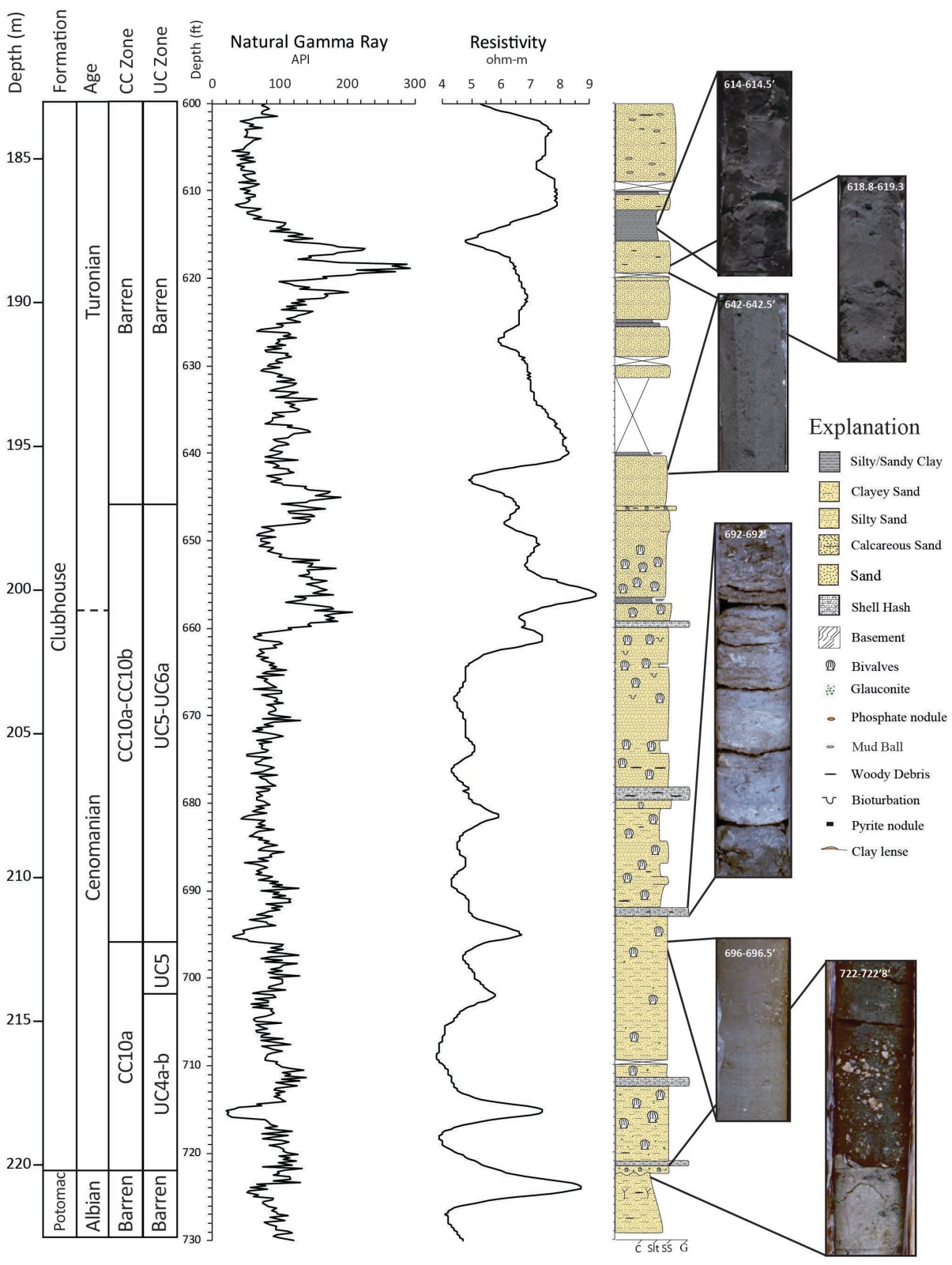

*Figure 2.* Stratigraphic column for Hope Plantation Core with CC and UC calcareous nannoplankton biozones, natural gamma ray and resistivity logs, and representative core images. C = clay; Slt = silt; SS = sand; G = gravel.

The Clubhouse Formation in the Hope Plantation Core (Figure 2) was penetrated between 174.3
m and 220.2 m below the surface. It is underlain by the floodplain paleosols of the Albian Potomac Group
(Thornburg, 2008) and is overlain by undifferentiated sands and muds questionably assigned to the Cape
Fear Formation (Weems et al., 2007). The Clubhouse Formation is primarily composed of clayey and
silty sands punctuated by a few discrete skeletal limestones. The whole unit coarsens upward from clayey
sands (from the base of the formation to about 210.0 m) to silty sands (from about 210.0 m to about 201.2
m) to more pure sands (from about 201.2 m to the top of the formation, although natural gamma ray peaks
suggest the inclusion of some clay in parts of this interval). This upper change corresponds with a clear
change in gamma ray log response that characterizes most of the informal marginal marine upper
member.
The lower marine informal member extends from the base of the Clubhouse Formation to the
highest common occurrence of bivalves and calcareous nannoplankton, around 196.9 m. Glauconite
occurs from the base of the informal marine unit up to about 211.2 m. Four decimeter-scale skeletal
limestones composed of broken bivalves occur roughly evenly spaced through this informal member.
Widely scattered woody debris is found between 210.6 m and 206.0 m. Definite bioturbation is rare but is
evident between 203.6 and 201.2 m, just below the shift in lithology from silty sand to cleaner sand.
Bivalves occur throughout the informal marine member in varying abundance. The marginal marine
upper informal member of the Clubhouse Formation is characterized by massive sand interbedded with
variably thick beds of massive silty clay, an increasing abundance of woody debris above 189.0 m, and
the occurrence of cm-scale mud balls above 185.6 m. A single thin bed containing bivalves occurs at
196.9 m. Given the more terrestrial features, cleaner sands, and thin clay interbeds of the upper informal
member of the Clubhouse Formation we suggest that these sediments were deposited in a non-marine or
marginal marine environment such as a distributary mouth bar or interdistributary bay system in the upper
part of the Clubhouse Formation.

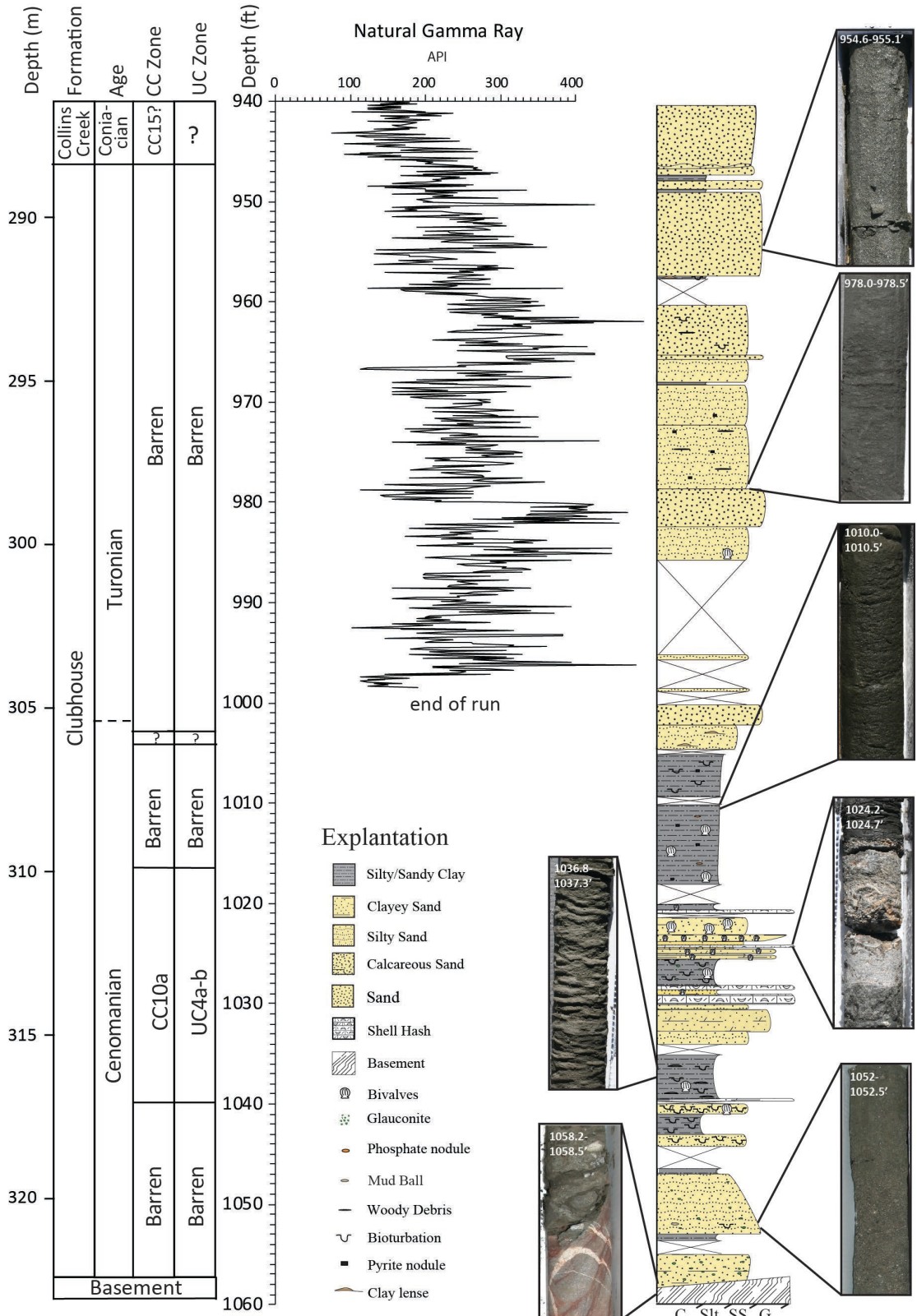

**Figure 3.** *Stratigraphic column for the Smith Elementary School core, with CC and UC calcareous nannoplankton biozones, natural gamma ray log, and representative core images.*

In the Smith Elementary core (Figure 3), the Clubhouse Formation occurs between 288.5 m and
322.7 m depth. Its basal contact with underlying gneiss is marked by a fault, with an angular contact
(~45° to vertical in the core) and slickensides (Weems et al., 2007). This fault is overlain by ~15 cm thick
interval of dolomitic sand. The lithology of the Clubhouse Formation in the Smith Elementary core is
overall more fine-grained than that of the Hope Plantation, with a lower fining-upwards interval, muds
and limestones in the middle, and a coarsening upward interval that extends to the unconformable upper
contact with the Santonian marginal marine Collins Creek Formation.
The lower marine informal member of the Clubhouse Formation in the Smith Elementary core
(322.7-~305.0 m) contains a more varied lithology than that of the Hope Plantation core. The basal
interval in this member is a 2.6 m thick package of massive, coarsening upward and then fining upward,
clayey to silty, glauconite-bearing sandstone separated by a thin silty claystone above a ~35 cm core gap.
Coring gaps of this scale are more common in the Smith Elementary core and are associated with the
contacts between sand and clay intervals. A single burrow occurs in the upper sandstone bed, and
glauconite decreases upsection. The overlying interval is composed of 2.8 m of bioturbated clay and silty
clay, with two ~30 cm thick silty sandstones with abundant burrows and rare bivalves. The upper silty
claystone contains thin clay lenses. This claystone is overlain by a 4.0 m thick interval of interbedded
silty- to clayey sandstone, skeletal limestones composed of broken bivalve debris, including one which
has been dolomitized, and a thick (~ 80 cm) bioturbated silty claystone containing glauconite and bivalve
shells. This in turn is overlain by a 5.2 m thick silty claystone with planar bedding, phosphate nodules,
pyrite, and bivalve shells. The lower 3.4 m of this claystone is laminated with no visible bioturbation.
Overall this interval represents a fining upward sequence from sand to sandy silt to silty clay; the sandy
clay contains thin discrete beds of coarser material, include shell hash, possibly indicating deposition
above storm wave base before deepening to uniform silty clay representing deposition on the shelf below
storm wave base at the top of the informal marine member.
The upper marginal marine informal member of the Clubhouse Formation in the Smith
Elementary core (~305.0-288.5 m) is composed of meter-scale beds of silty to well-sorted sandstone
which generally become coarser up section, interbedded with centimeter scale beds of claystone. Some
beds contain woody debris and pyrite. A single bivalve occurs near the very base of the member, and a
few discrete burrows are observed between 294 and 292 m. Flaser bedding occurs in a clay bed at 291.7
m. The overall coarse-grained nature of these beds, and the alternating terrestrial and marine indicators
lead us to interpret this interval as being marginal marine, perhaps representing distributary mouth bars.
The overlying contact with the Collins Creek Formation is marked by a readily observable unconformity.
**4.2 Biostratigraphy**
Calcareous nannofossil assemblages are prevalent in the Hope Plantation core (Figure 4), with
abundances ranging from rare to common and preservation from good to poor; the top of the Clubhouse
Formation is barren (196.8-185.5 m) (Self-Trail et al., 2021). The base of the Clubhouse Formation is
placed in the late Cenomanian Zone UC4a-b of Burnett (1998) and Zone CC10a of Sissinghi (1977)
based on the presence *of Lithraphidites acutus*, whose highest occurrence (HO) at 214.0 m marks the top
of Zone UC4b. The absence of *Cretarhabdus loriei*, whose HO marks the top of UC4a, could be due to
ecological exclusion from inshore environments, and thus sediments in this interval are lumped together
into a combined zone (UC4a-b).  A condensed (or truncated) interval from the HO of *L. acutus* to the HO
of *Helenea chiastia* at 212.9 m is placed in Zone UC5 (undifferentiated) and is latest Cenomanian in age.
It is unclear from nannoplankton data alone whether the HO of *H. chiastia* is the true extinction of this
taxon (and thus this level marks the latest Cenomanian) or if this absence of this species above the level is
the result of poor preservation and/or ecological exclusion from the inner shelf as increased terrigenous
flux made the waters less welcoming to marine nannoplankton. We favor the latter explanation, because
the sample immediately above the highest *H. chiastia* is barren, and marks the beginning of an interval
characterized by poor preservation and locally barren samples. This interval, from 212.1-197.0 m, is
placed in zones UC5-UC6a and CC10a-CC10b based on the absence of both *H. chiastia* and *Eprolithus*
*moratus*, whose lowest occurrence (LO) defines the base of Zone UC6b. The Cenomanian/Turonian
boundary is placed at 200.3 m based on carbon isotope data (see section 4.3.1, below).

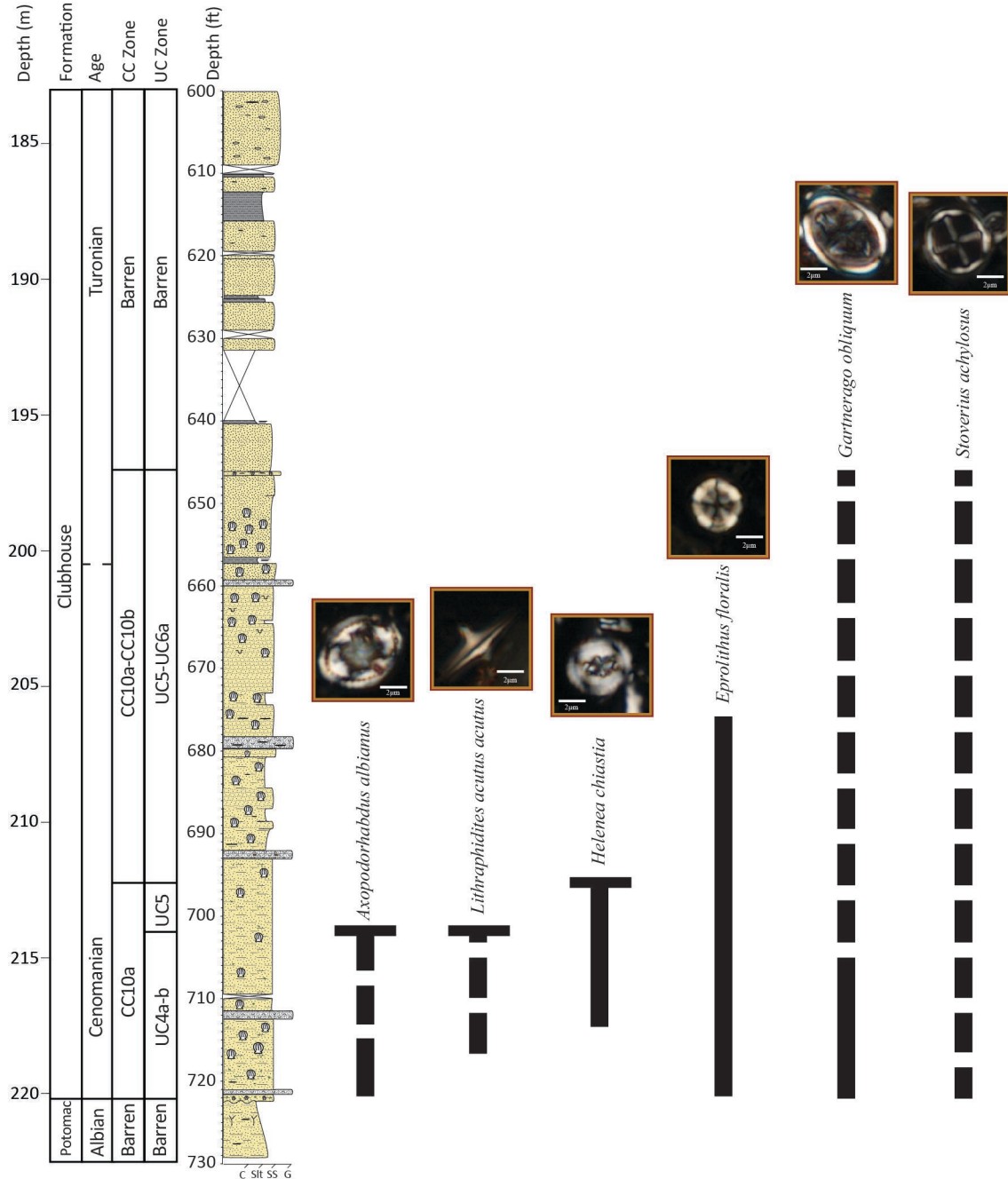

***Figure 4.*** *Ranges of key calcareous nannoplankton species in the Hope Plantation Core. Dashed lines indicate*
*sporadic occurrence.*

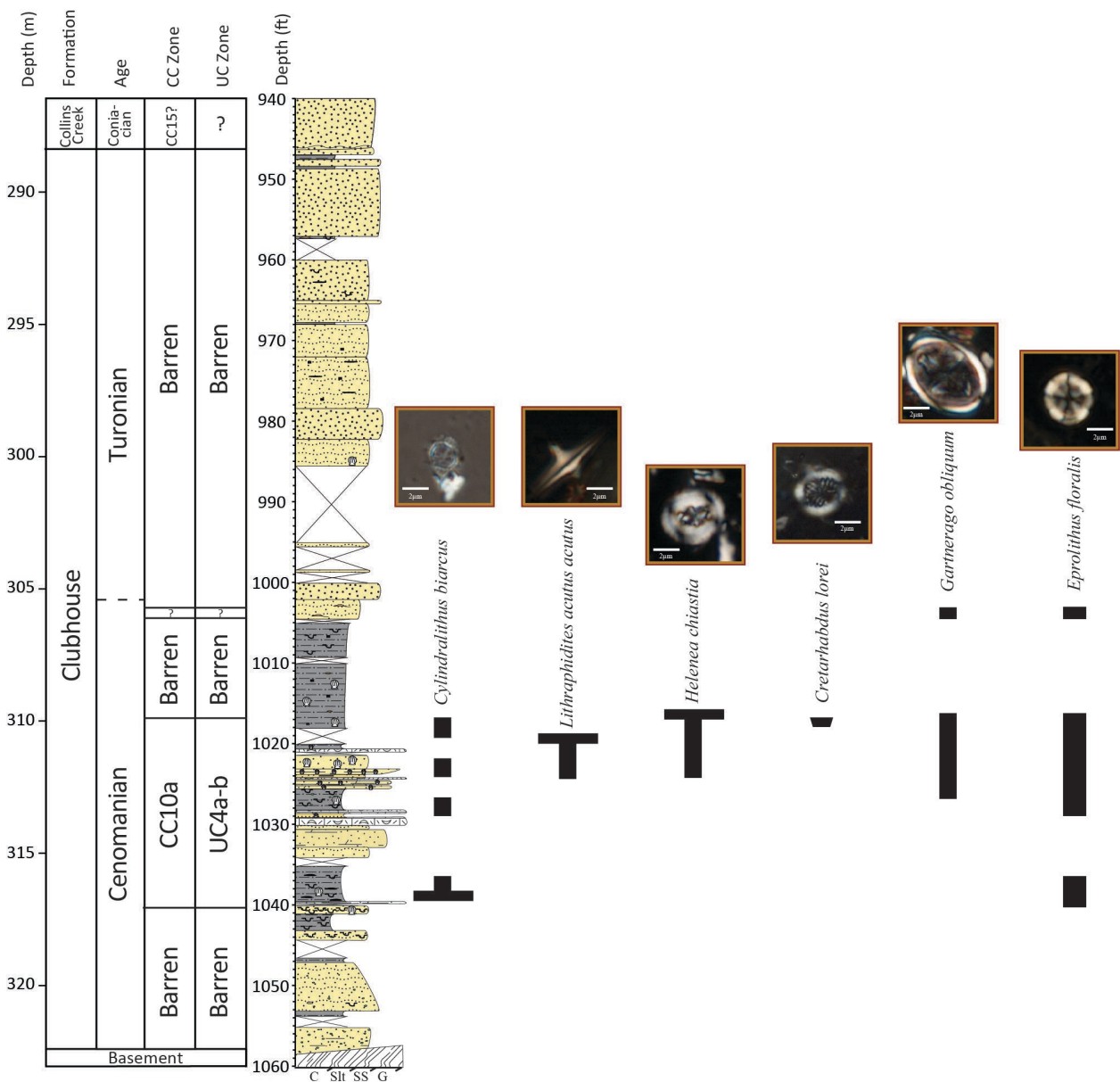

**Figure 5.** *Ranges of key calcareous nannoplankton species in the Smith Elementary School core. Dashed lines indicate sporadic occurrences.*

Calcareous microfossils are only sporadically present in the Smith Elementary School sediments (Self-Trail et al., 2021) (Figure 5). Even though the presence of glauconite, burrowing, fish debris and scattered shell fragments indicates deposition in a marine environment, intervals barren of calcareous nannoplankton are common and extensive, from 322.5-317.3 m and 309.9-290.4 m (Figure 5). The presence of *Cylindralithus biarcus* at 316.4 m, the HO of *L. acutus* at 310.9 m, and the HOs of *H. chiastia*

and *C. loriei* at 310.2 m place this interval in the late Cenomanian calcareous nannofossil Zone UC4a-
b/Zone CC10a. The rare occurrence of poorly preserved calcareous nannofossils at 305.9 m suggests
continued placement in the Cenomanian or Turonian, but no diagnostic species were recovered, and thus
the Cenomanian/Turonian boundary must once again be placed using carbon isotopes at 305.4 m (see
section 4.3.1, below). An unconformity at the top of the Clubhouse Formation (288.4 m) corresponds to a
change from a barren interval below to a Santonian assemblage of calcareous nannofossils above.
All samples examined for planktic and benthic foraminifera were barren of whole specimens.
This is unlikely to be the result of anoxia at the time of deposition, as this would not explain the lack of
planktic foraminifera which occupied a mixed layer habitat similar to the nannoplankton observed in the
same interval. A few samples contained very rare fragments of both planktic and benthic foraminifera,
indicating that foraminifera were present in these sections but that they were subsequently dissolved,
either in situ or in the 17 years since the cores were drilled. This may be due in part to the relatively
organic-rich nature of the sediments and to the presence of pyrite, both of which have been found to result
in dissolution of calcareous microfossils in cored sediments of the Atlantic Coastal Plain (Self-Trail and
Seefelt, 2005; Seefelt et al., 2015). However, the well-documented occurrence of planktic and benthic
foraminifera in more distal coastal plain cores (e.g., Valentine, 1982, 1984; Zarra, 1989; Gohn, 1992)
bodes well for future micropaleontological studies in this region.
**4.3 Geochemistry**
**4.3.1 Carbon Isotopes**
Organic carbon isotope ($\delta^{13}C$) data (Figure 6) in each core show clear positive excursions
associated with OAE2 in the marine interval of the Clubhouse Formation. Both isotope records display a
~2‰ positive shift with the classic A-B-C structure of OAE2, with an initial excursion (A), a brief
recovery followed by a second peak (B) and a longer plateau with a small peak (C) first described by Pratt
and Threlkeld (1984) in the US Western Interior Seaway. The Hope Plantation core, which is
characterized by coarser grains and a more proximal environment, has a more expanded OAE2 interval (~
17.4 m) compared to the somewhat more distal Smith Elementary Core (~ 10.4 m). We compare the
expanded Hope Plantation carbon isotope record to other representative North American OAE2 records
from the Western Interior Sea and the Gulf of Mexico and Atlantic coastal plains in Figure 7. The
termination of the OAE2 carbon isotope excursion roughly corresponds with the Cenomanian–Turonian
boundary (e.g., Kennedy et al., 2005) and has been used to define that level in our cores.
**4.3.2 Total Organic Carbon**
Total organic carbon data (Figure 6) reveals relatively low enrichment in organic carbon in the
Hope Plantation core, generally <1 weight percent (wt%) TOC except for a few discrete peaks associated
with woody debris. Average values are slightly higher during OAE2 (~0.6 wt%) compared to background
levels in the overlying interval (~0.4 wt%) but just barely. Values are slightly higher overall in the Smith
Elementary School core, particularly during OAE2, where the upper part of the event averages about 1.0
wt% TOC.
**4.3.3 Organic Carbon/Nitrogen Ratios**
The ratio of total organic carbon to total nitrogen is a common proxy for the relative contributions
of algae and land plants to sedimentary organic matter (e.g., Meyers, 1994, 1997, 2003). Due to
differences in their composition (e.g., the abundance of cellulose in land plants) vascular plants tend to
have C/N ratios of 20 or greater, while algae have C/N ratios of 4-10 (Meyers, 1994). Changes in C/N
ratio in marine settings therefore reflect changes in the relative contribution of terrigenous organic matter
to offshore areas. C/N can thus be used to reconstruct changes in the hydrologic cycle, with increased C/N
ratios indicating a higher flux of terrestrial organic matter due to enhanced weathering (Meyers, 2003).
Sediments with low TOC (<0.3 wt%) can cause problems for C/N interpretations because in such settings
the proportion of inorganic nitrogen can be high enough to artificially depress the data, suggesting more
marine organic matter than is really there (Meyers, 1997); our data is consistently above 0.5 wt% TOC so
this is not a concern (see section 4.3.2, above).

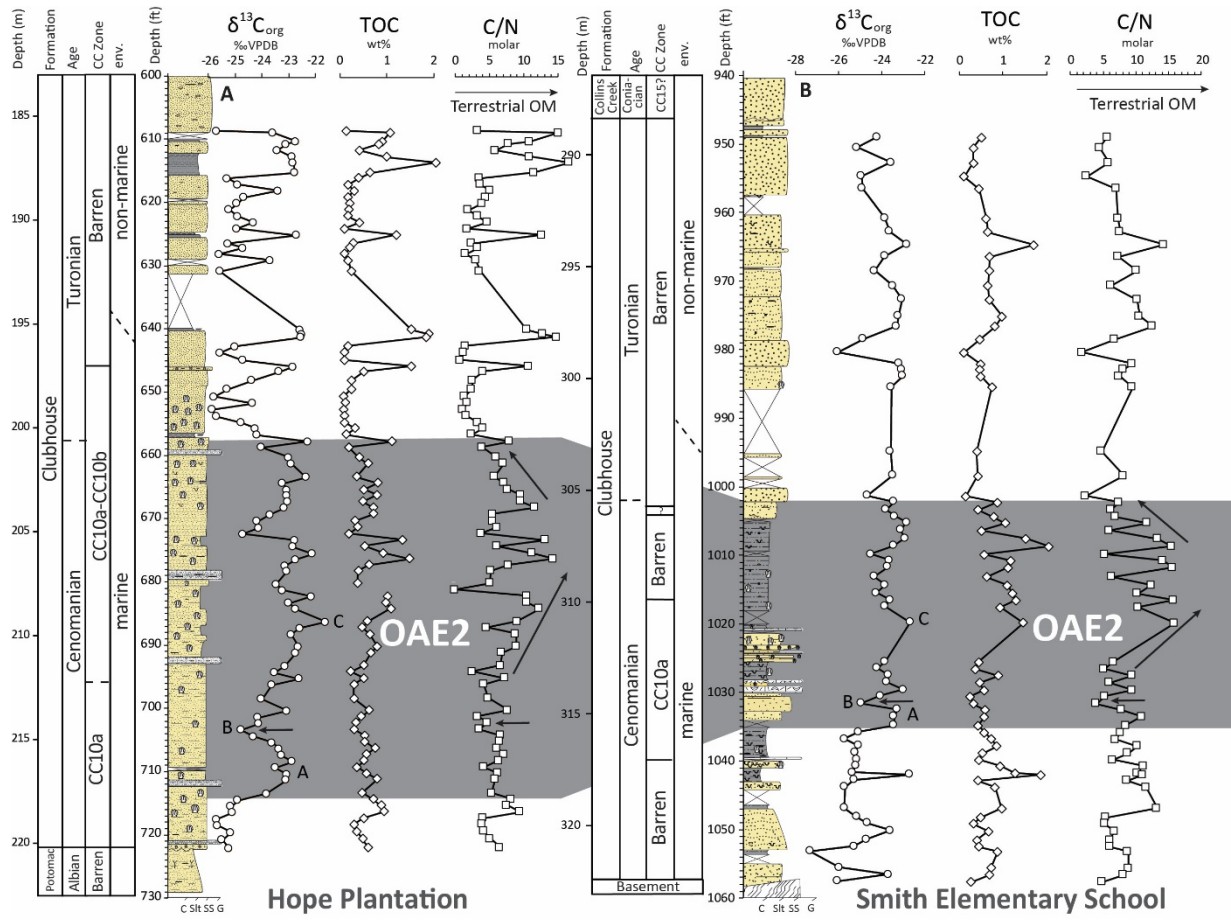

*Figure 6. Geochemical data from the Hope Plantation (left) and Smith Elementary School (right) cores plotted against stratigraphic columns for each. Grey shaded area represents the OAE2 interval in each core. Letters A-B-C labels on carbon isotope ($\delta^{13}C$) curve correspond to named points of the OAE carbon isotope excursion. TOC = total organic carbon; C/N = carbon/nitrogen ratio. Arrows indicate brief reduction in C/N ratio coincident with the Plenus isotope excursion ("B" on the $\delta^{13}C$ plot) and broad increase in values during the main part of the $\delta^{13}C$ excursion. Note slight change in depth scale between the two cores, as the studied interval in Smith Elementary is 10 ft (3.1 m) thicker than Hope Plantation.*

C/N ratios in both cores are elevated during OAE2, indicating enhanced contribution of terrestrial
organic matter driven by a strengthened hydrologic cycle (Figure 6). In the Hope Plantation core, C/N
ratios increased from an average of 5.5 prior to OAE2 to 7.1 during the event, with higher values later in
the event, peaking around 14.4. Average values dropped back to 5.5 after OAE2, including occasional
peaks reflecting the inclusion of woody plant debris visible in the core. In the Smith Elementary School
core, C/N ratios increased from 8.4 before OAE2 to 9.53 during OAE2, with peak values (up to 16.0)
again occurring later in the event. Post-OAE2 C/N values at Smith Elementary are less noisy than those at
Hope Plantation, and average 7.4.

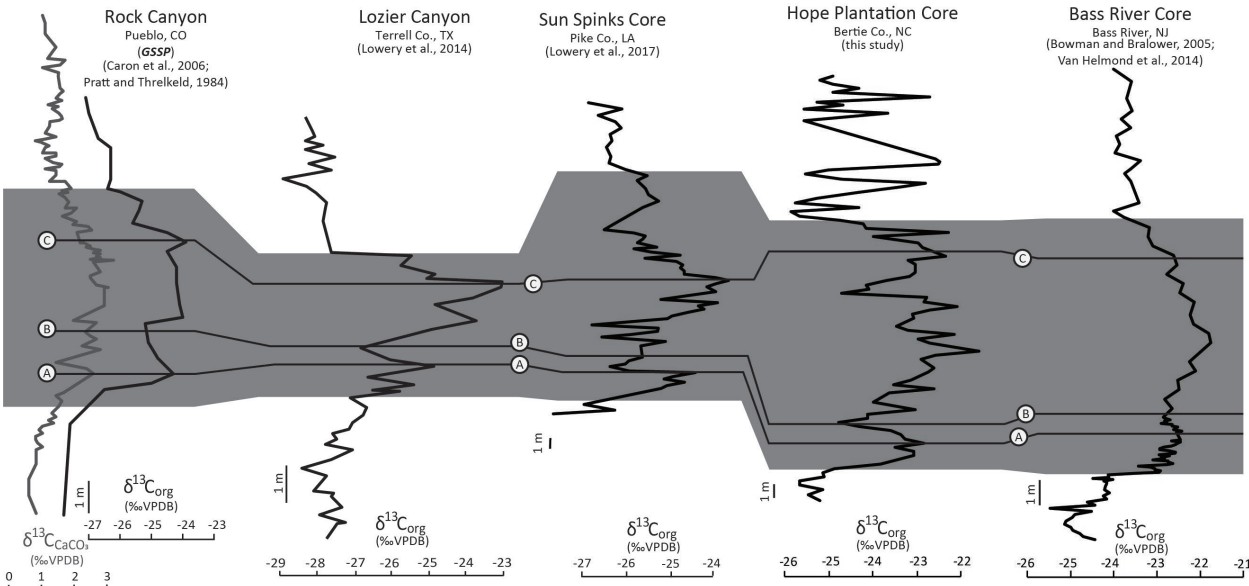


***Figure 7.*** *Comparison of North American carbon isotope curves from the Cenomanian-Turonian GSSP at Rock*
*Canyon in Pueblo, CO; Lozier Canyon in Terrell Co., TX, near the transition from the Western Interior Seaway to*
*the Gulf of Mexico; the Sun Spinks core in Pike Co., LA on the Gulf Coastal Plain, the Hope Plantation Core from*
*Bertie Co., NC on the Atlantic Coastal Plain, and Bass River core from Bass River, NJ on the Atlantic Coastal*
*Plain. The Rock Canyon record includes both bulk carbonate (grey line) and organic carbon (black line) carbon*
*isotopes; all other sites only show organic carbon isotopes. The position of the A-B-C peaks are traced between the*
*cores. Grey bar shows the extent of OAE2 in each core.*
**5 Discussion**
**5.1 Enhanced Hydrologic Cycle During OAE2**
Our data indicate a strengthened hydrologic cycle in southeastern North America preceding the
start of OAE2 and continuing through the event, in agreement from the data from van Helmond et al.
(2014) some 500 km to the north. Palynological data from New Jersey agree with our bulk geochemical
data in showing highest terrigenous flux during the latter part of the OAE2 isotope excursion. The pre-
event increase in terrigenous flux is an interesting parallel to records of pre-event global oxygen
drawdown based on thallium isotopes (Ostrander et al., 2017), suggesting a link between weathering flux
and deoxygenation, likely via enhanced delivery of nutrients to the oceans. Additionally, a drop in C/N
ratio in both of our core records during the carbon isotope minimum referred to as the Plenus carbon
isotope excursion (O'Connor et al., 2020) indicate relatively drier conditions at this time, a phenomenon
also observed in New Jersey coincident with a decrease in temperatures (van Helmond et al., 2014). The
Plenus Cold Event was originally interpreted as a global cooling event (hence the name, e.g., Gale and
Christensen, 1996; Erbacher et al., 2005; Jarvis et al., 2011; Hasegawa et al., 2013; Gale, 2019).
However, more detailed comparisons of temperature and carbon isotope records from a wide range of
sites has demonstrated that the timing and magnitude of cooling varies significantly by location
(O'Connor et al., 2020). Our results agree with those of van Helmond et al. (2014) that the Plenus interval
resulted in a weaker hydrologic cycle and reduced terrigenous flux into the oceans, at least along the east
coast of North America.
**5.2 OAE2 on the eastern North American shelf**
The Smith Elementary School and Hope Plantation cores represent the second and third records
of OAE2 on the US Atlantic Coastal Plain. As such, they provide important insight into a surprisingly
understudied region. In the modern ocean, about 85% of organic carbon burial occurs along continental
margins (e.g., Burdige, 2007). A survey of all known OAE2 localities with a complete carbon isotope
excursion and TOC data by Owens et al. (2018) found that there is a significant amount of "missing"
organic carbon when reconstructed organic carbon burial is compared to "expected" carbon burial based
on carbon isotope data. This was based on 170 sites which, with some extrapolation, represent just 13%
of total Cenomanian–Turonian global ocean area, which meant that similar values had to be assumed for
the rest of the seafloor (Owens et al., 2018). OAE2 is perhaps the best studied event of the Cretaceous,
but these results suggest a clear need for additional sites to better constrain paleoceanographic and
paleoenvironmental changes during this event. By adding additional OAE2 sites on the Atlantic Coastal
Plain our results help to constrain the contribution of these areas to global carbon burial.

Van Helmond et al. (2014) point out that TOC is lower in the Bass River core than other OAE2

sections in the North Atlantic region, but our results indicate that Bass River is about average for inner
continental shelf deposits (Figure 8). Average TOC during OAE2 at Bass River is 1.1 wt% (van
Helmond, 2014); this is slightly higher than Smith Elementary (0.83 wt%) and Hope Plantation (0.63
wt%) and slightly lower than the next closest published shelf site to the southwest, the Sun Spinks core in
Mississippi (1.4 wt %, Lowery et al., 2017). Sequence stratigraphic analysis of Cenomanian/Turonian
sediments of the Clubhouse and Bass River Formations show that these sediments represent maximum
sea level rise across the boundary on the Atlantic Coastal Plain (Miller et al., 2004; Aleman Gonzalez et
al., 2020). The location of the Hope Plantation core (lowest TOC values) higher on the inner paleoshelf
relative to Smith Elementary School and Bass River (higher TOC values) suggests that TOC wt% on the
shelf during OAE2 was, at least in part, a function of paleodepth.  To be sure, these TOC values are
certainly lower than values found offshore in the open ocean or along upwelling margins in the eastern
proto-North Atlantic. For example, Deep Sea Drilling Project Site 603, on the lower continental rise
directly offshore of North Carolina, has an average TOC of 5.4 wt % during OAE2 (Kuypers et al., 2004),
while the upwelling-prone region at Tarfaya, Morocco has an average TOC of 8.0 wt% (Kolonic et al.,

2005).

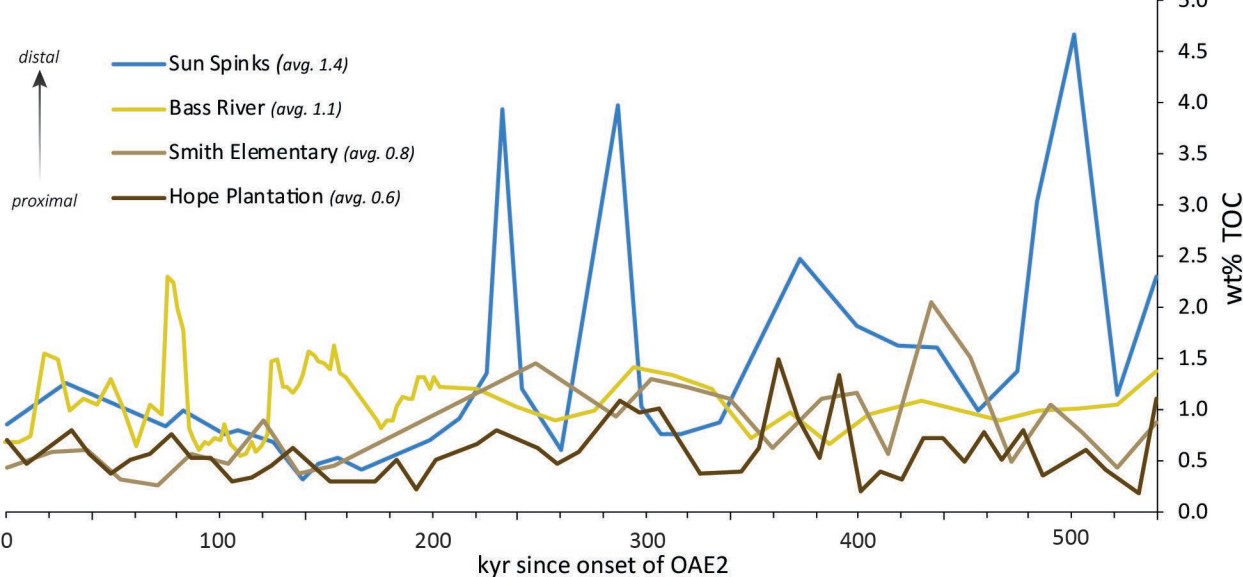


**Figure 8.** *Comparison of measured wt% TOC for the duration of OAE2 for the Sun Spinks core in Mississippi, Bass*
*River core in New Jersey, and the Smith Elementary School and Hope Plantation cores in North Carolina. Age*
*model based on the thickness of the OAE carbon isotope excursion and the orbitally-tuned duration of OAE2 at the*
*Global Stratotype Section and Point in Pueblo, CO (Sageman et al., 2006; Meyers et al., 2012) of 540 kyr, assuming*
*a constant sedimentation rate.*

Sedimentation rate also plays an important role in organic carbon accumulation. While we don't

have dry bulk density measurements from these cores to calculate mass accumulation rates, we can
approximate using reasonable values for organic-rich silicilastic rocks (2.4 g/cm$^3$, following Owens et al.,
2018). We can determine the average sedimentation rate during the event using the observed thickness of
the OAE2 carbon isotope excursion in each core and the orbitally-tuned duration of OAE2 at the Global
Stratotype Section and Point in Pueblo, CO (Sageman et al., 2006; Meyers et al., 2012) of 540 kyr. A
constant sedimentation rate on the shelf during OAE2 is almost certainly an oversimplification but it is
sufficient for our purpose of comparing general trends between these cores. Using these values we find
organic carbon mass accumulation rates (OC MAR) during OAE2 average 0.05 g/cm$^2$/kyr at Hope
Plantation, 0.04 g/cm$^2$/kyr at Smith Elementary School, 0.06 g/cm$^2$/kyr at Bass River, and 0.11 g/cm$^2$/kyr
at Spinks. For comparison, the same method indicates OC MAR rates of 0.29 g/cm$^2$/kyr at DSDP Site 603
and 2.84 g/cm$^2$/kyr at Tarfaya (Owens et al., 2018). Owens et al. (2018) found an average OC MAR on
shelf sites during OAE2 of 0.11 g/cm$^2$/kyr, which means the inner shelf sites on the east coast of North
America are below the global average during this event.
These data suggest a relationship with depth on the shelf and TOC deposition during OAE2. If
we arrange the sites by depth (Figure 8) we see the lowest average TOC values at Hope Plantation (0.63
wt%), the most proximal site; values are slightly higher at Smith Elementary (0.83 wt%) which appears to
represent an outer estuary or inner shelf environment, and higher still at Bass River (1.1 wt%) which was
inner to middle neritic (Miller et al., 2004). Estimates of organic carbon mass accumulation rates suggest
all three of these inner shelf sites are very similar, ranging from 0.4-0.6 g/cm$^2$ kyr. Average TOC is even
higher in the Spinks Core (1.4 wt%, or 0.11 g/cm$^2$ kyr), which represents inner to middle neritic depths
during the latter part of OAE2 (Lowery et al., 2017). This suggests the possibility of even higher values
on more distal parts of the shelf, and highlights the need for a true depth transect (as opposed to four cores
from three states) to better understand that variability and better constrain organic carbon burial in this
important environment during OAEs.
**6 Conclusions**
Calcareous nannoplankton biostratigraphy shows that positive carbon isotope excursions in two
cores on the Atlantic Coastal Plain in North Carolina are associated with the Cenomanian–Turonian
OAE2. C/N ratios in both cores indicate an increase in the proportion of land plants delivered to these
offshore sites during, indicating a strengthened hydrologic cycle causing increased terrigenous flux
beginning slightly before OAE2 and continuing through the whole event. This agrees with palynology-
based observations from the Bass River core located ~500 km to the north (van Helmond et al., 2014).
We therefore conclude that these changes reflect increased precipitation and weathering across eastern
North America during OAE2, feeding nutrients onto the shelf and into the proto-North Atlantic, and
likely contributing to the widespread black shale deposition in the deep basin. These cores are the second
and third records of OAE2, to our knowledge, on the coastal plain of eastern North America and,
combined with the first (Bowman and Bralower, 2005; van Helmond et al., 2014), show relatively low
average TOC values (~0.6 - 1.1 wt%) on the inner shelf during this event, while suggesting a trend of
increasing values with depth, highlighting the need for more cores in this region from middle and outer
shelf depths.
**Data Availability Statement**

Total organic carbon, total nitrogen, organic carbon isotope, geophysical and calcareous

nannofossil occurrence data are published (Self-Trail et al., 2021) and available for download as a USGS
Data Release at https://doi.org/10.5066/P9V0U1NF.
**Author Contribution**
CL and JS conceived of the study and sampled the cores. JS sat the wells in 2004 and 2005 and helped
describe the cores. CB conducted bulk organic carbon/nitrogen and organic carbon isotope measurements.
JS conducted calcareous nannoplankton biostratigraphy. CL supervised foraminifer analysis. CL prepared
the manuscript with contributions from JS and CB.
**Competing Interests Statement**
The authors declare that they have no conflicts of interest.
**Acknowledgements**
We are grateful to the drillers and personnel of the USGS for taking these cores, to the staff of the North
Carolina Geological Survey for maintaining the cores and making them accessible for sampling, to Lara
Yagodzinski for her help sampling the cores, to Kate Gilbreath for her help preparing samples for
micropaleontological analysis, and to Ellen Seefelt for her assistance preparing samples for nannofossil
analysis and for shipping material. Core box photographs courtesy of USGS and NCGS. We acknowledge
the people, up to 200 at a time, who were held as slaves at Hope Plantation between 1748 and 1865.  Any
use of trade, firm, or product names is for descriptive purposes only and does not imply endorsement by
the U.S. Government.

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
