# Peer review of "Enhanced Terrestrial Runoff during Oceanic Anoxic Event 2 on the North Carolina Coastal Plain, USA"

_Climate of the Past, 2021_

## Author Comment (AC2)

**Rock Canyon**
Pueblo, CO
(***GSSP***)
(Caron et al., 2006;
Pratt and Threlkeld, 1984)

**Lozier Canyon**
Terrell Co., TX
(Lowery et al., 2014)

**Sun Spinks Core**
Pike Co., LA
(Lowery et al., 2017)

**Hope Plantation Core**
Bertie Co., NC
(this study)

**Bass River Core**
Bass River, NJ
(Bowman and Bralower, 2005;
Van Helmond et al., 2014)

$\delta^{13}C_{org}$
(‰VPDB)

-27   -26   -25   -24   -23

$\delta^{13}C_{CaCO_3}$
(‰VPDB)

0   1   2   3

$\delta^{13}C_{org}$
(‰VPDB)

-29   -28   -27   -26   -25   -24   -23

$\delta^{13}C_{org}$
(‰VPDB)

-27   -26   -25   -24

$\delta^{13}C_{org}$
(‰VPDB)

-26   -25   -24   -23   -22

$\delta^{13}C_{org}$
(‰VPDB)

-26   -25   -24   -23   -22   -21

1 m

---

## Author Response (AR1)

Dear Dr. Sluijs,

On behalf of my co-authors I'd like to thank you for facilitating these two helpful reviews of our manuscript. This is my first time leading a paper in Climate of the Past and I apologize for being slow with responses to the public comments, finalizing my comments after the review period ended, etc. It's definitely not what I'm used to with reviews but I'd say that it was definitely a positive experience. Our response to the reviews (very similar to that posted publicly, just now written in a different tense) is below. We completely agree that the goal should be clarity for readers and have endeavored to make those changes wherever possible. The main change to this version of the manuscript is the inclusion of a new figure showing how our new OAE2 carbon isotope curves fit with others across N. America. Replies to each comment can be found below.

Best,

Chris Lowery

**Reviewer #1**

*This paper presents an integrated bio- and chemostratigraphic profile of the mid-Cretaceous Atlantic Coastal Plain, USA, along with a geochemical perspective on terrigenous input into the basin.*

*This paper is very well written, and the scientific methods and findings are sound. I believe that the data presented here are valuable to the scientific community and will be of interest to the readers of Climate of the Past. I suggest only minor revisions.*

We are happy to hear that the reviewer found the paper well-written and worthy of publication in Climate of the Past with minor revisions, and we appreciate their helpful comments on the text.

*Minor points:*

*-En dashes should be used instead of hyphens when indicating 'to'. e.g., Cenomanian–Turonian.*

fixed

*-Line 141: what kind of work? Biostratigraphy?*

Yes. This has been clarified.

*-Line 172: what molarity was the acid?*

The acid is 1M HCl

*-Line 195, 324, 385: please avoid using terms such as 'this' or 'these' without qualifying. Check throughout*

I don't see an unqualified "this/these" at line 195 but we have fixed the other two and tried to clarify these terms throughout the text.

*-Line 200: 201.2 m*

fixed

*-O'Connor et al. 2019 should be 2020*

fixed

*-Line 355-360: this interpretation is incorrect. O'Connor et al. state that there was no drawdown of CO2. The original Barclay et al. data indicate an increase in CO2, and this interpretation is supported by O'Connor et al.*

To avoid lengthening the paper with a detailed discussion of the evolving understanding of the Plenus event (especially the role of CO2 drawdown, which, as the reviewer states, is not supported by the evidence), which would have eventually ended with a refutation of the CO2 drawdown anyway, we have simply deleted the reference to reduced CO2 in this section, and limit our discussion to the correlation to observed cooling and reduced terrigenous flux in the Bass River core by van Helmond et al.

**Reviewer #2**

*This study presents two new sedimentary records of the Atlantic Coastal Plain of North Carolina, USA, covering Oceanic Anoxic Event 2 (OAE2), one of the largest carbon cycle perturbations of the Mesozoic. The stratigraphic framework is based on calcareous nannofossils and (bulk) organic carbon stratigraphy. Sedimentary TOC is used to assess inner shelf organic carbon burial and C/N ratios are used to reconstruct terrestrial input, indirectly linking to the local/regional hydrological cycle. TOC content is relatively low and C/N ratios are relatively high for the OAE-2 interval pointing to low inner shelf carbon burial and increased terrestrial input linked to an enhanced hydrological cycle, in accordance with previous studies.*

*This paper is well written and the methods used are robust. While the findings of this study are not leading to major new insights the introduction of two new mid-Cretaceous sedimentary records of the Atlantic Coastal Plain of North Carolina is valuable and warrants publication in this journal. I do have some comments and suggestions that the authors might want to use for improvement of their manuscript.*

We glad the reviewer found our paper well-written and our results valuable (we agree that they don't represent a paradigm shift but feel that slowly building more detailed understanding of major events is an equally important, if less glamorous, part of the scientific process). We grateful to the reviewer for their thoughtful comments and suggestions, which we respond to in detail below. In particular, we really like the idea of a new figure placing our cores in the context of some classic reference sections, and have thus created a new Figure 6 which we call out in the carbon isotope results section.

*Comments:*

*Calcareous nannofossils are used as a biostratigraphic tool in this study, but I was wondering whether their assemblages are also providing insight into the paleoenvironmental conditions on the Atlantic Coastal Plain during OAE2? If this is the case, if might be worth to add a brief section on this.*

In general, nannofossils (especially from the shelf) can give clues regarding the paleoenvironment. Relative abundances of certain species can indicate increased fertility, cold vs. warm, and possibly salinity.

Having said that, we think the nannofossil assemblages in both Hope and Smith were affected by dissolution (both in situ and probably post-coring) and are therefore not very reliable to be used for any paleobiogeographic analyses. They're good enough for biostratigraphy (most of the markers are robust), but we'd be really cautious about using them for anything else.

*It would be helpful for potential follow-up work to present these new records in a more global stratigraphic framework. Perhaps an overview figure (potentially as a supplemental figure) including some key d13C curves (e.g., Eastbourne, Pueblo), the proximate Bass River core and the new d13C curves could be included? Such figure could also help to directly compare these new records with the two already existing records (Bass River and Sun Spinks) brought forward for discussion.*

This is a great idea, and we've created a new Figure 6 to show these relationships. For space reasons we had to leave Eastbourne out because we wanted to focus on key North American sections, which are more relevant (in our opinion) to the present work.

*Minor Comments:*

*Line 14: I would rephrase "North Atlantic Basin" to "proto-North Atlantic basin." Idem for lines 376 and 386.*

Changed.

*Line 37: I would add the approximate age of the event, i.e., ~ 94 Ma.*

added

*Line 51: In the van Helmond et al. (2014) study the interval with the highest terrestrial input correlates with the cold event during OAE2. I think the inference by van Helmond et al. of an enhanced hydrological cycle is based mainly on the relatively high abundances of a group of dinoflagellate cysts associated with low sea-surface salinities during the warmest phases. This actually brings me back to my comment above, do the calcareous nannofossil assemblage provide any information on for example salinity?*

As stated above, the likelihood of dissolution of some more delicate species of nannoplankton makes us hesitant to draw any conclusions from the overall population.

*Line 61: "therefore" seems to be unnecessary in this sentence.*

It definitely is. Deleted.

*Line 67: Important to mention that these are "molar ratios." Unfortunately I have seen that sometimes weight percentages are used instead.*

mentioned

*Line 145: was this core also drilled as a stratigraphic test for Atlantic Coastal Plain Aquifers?*

Yes it was they were part of the same project. You're right that we should mention that here too for consistency.

*Line 170: I would first mention the elemental composition and then the isotopes, that is the sequence you use for the remainder of the manuscript.*

Good idea, we've changed it.

*Line 172: please correct "there was no…"*

corrected

Line 178: Are these absolute or relative uncertainties? Were duplo's or triplo's ran? What was the average analytical uncertainty of those?

These are absolute uncertainties. Every 5$^{th}$ sample was run in duplicate which was to confirm the precision and reliability of the measurements while the check standard was run in triplicate every 20 samples to ensure the accuracy of the results. The standard deviation of the composition results from the duplicates was within the range of the absolute uncertainty quoted. The uncertainty itself being determined from the range of results reported from the check standard used throughout the analyses. We have added this information to this section.

*Line 180: idem as for line 178*

The same as above, absolute in terms of uncertainty. The samples were run in duplicate every 5 samples for compositional and isotopic assessment. These duplicates are what define the precision/reproducibility of the data. Unlike compositional results all isotopic data is relative and therefore, the start of any analytical run requires a series of known standards to be run, covering the potential range of the unknown isotopic results, and these are also used to determine the overall accuracy of the data being produced. A check standard is run every 20 samples to ensure consistency and to monitor for any drift. As with the compositional results the overall uncertainty is determined from the range of results across the standards analyzed. We've added this information to the text as well.

*Lines 188-192: Is it possible to estimate the paleo-water depth?*

We would hesitate to put numbers on it without additional data. This is where the lack of benthic foraminifera really hurt, as they are perfect for this kind of question. I would say that depths probably range from like 10 to 50 m, but that's not based on much more than observations of the sediments and general understanding of inner shelf environments.

*Line 200: What do the authors mean with "cleaner sand"*

More pure sand (i.e., with less mud and silt mixed in). We've changed "clean" to "pure" to make sure our meaning is clear.

*Line 239: "Core" should be "core"*

Fixed.

*Line 253: What environmental conditions?*

Probably ecological exclusion of open ocean taxa from nearshore waters, which we now state in the text for clarity.

*Line 283: Could (seasonally) low oxygen conditions be an alternative explanation for the almost absence of benthic foraminifera? Can the authors give a rough estimate of the bottom water redox conditions at both sites based on their study?*

Probably not. We are very familiar with low-oxygen assemblages of benthic foraminifera during OAEs (see Lowery et al., 2014, 2017a, 2017b, and 2018) and unless the seafloor was entirely anoxic we would expect to see at least a few individuals from low-oxygen tolerant genera. Such sustained anoxia is very unlikely in shallow coastal waters, particularly given the observation of burrows in some intervals, which make most or all of the interval appear to be at least somewhat oxygenated. If anoxia were driving the lack of benthic foraminifera then we'd expect to see intervals with some benthic foraminifera and burrows, and other laminated intervals with no benthic foraminifera. Additionally, we'd expect to see planktic foraminifera throughout, as those occupy a mixed layer habitat that is unaffected by benthic anoxia. The lack of all foraminifera indicates that the issue is one of preservation and paleoenvironmental conditions. We now briefly mention that anoxia is not a likely explanation for the lack of foraminifera at this point in the text.

*Lines 286-287: I wonder whether the TOC content in the Smith Elementary School and Hope Plantation is related to the absence of foraminifera. TOC in the Bass River core is higher, yet that core does contain plenty of foraminifera (Sugarman et al., 1999, JFR)*

Probably not, for the reasons discussed above. Other cores have even higher TOC during the event and still have plenty of benthic forams. And, this would not explain the lack of planktic foraminifera.

*Section 4.3.3: Maybe the authors can include a table with average, minimum, and maximum TOC and C/N prior to OAE, during OAE2 and post OAE2 to make the differences more prominently visable. Based on Figure 6 alone a reader might not be convinced of the shifts the authors describe.*

Average TOC values during OAE2 are included on Figure 7. Average C/N values are presented here:

| C/N | Hope | Smith |
|---|---|---|
| Peak during OAE2: | 14.4 | 16 |
| avg. pre-OAE2 | 5.86 | 8.35 |
| avg. OAE2 | 7.07 | 9.53 |
| avg. post-OAE2 | 5.46 | 7.43 |

It's probably easier to work these numbers into the text than to add a new table to the manuscript, so we've re-written this section to call out these numbers for each core.

*Line 346: please see my comment on Line 51*

Please see our response above.

*Line 377: How does this compare to modern values for these settings?*

We're not sure that a meaningful comparison can be made between particular locations on the Cretaceous continental shelf and their modern equivalents, given the major changes in ocean circulation, climate, and primary producers that have occurred in the intervening 94 million years.

*Line 398: "2.4 g/cm2" should be "2.4 g/cm3"*

Fixed.

*Figures:*

*Figure 1: Is there any evidence of major rivers flowing out on the Atlantic Coastal Plain? If so, it would be helpful to indicate their position.*

The short answer is that we are not aware of any studies showing the positions of major rivers flowing onto the Atlantic Coastal Plain during the Cenomanian-Turonian.

The longer answer is that there is some indication in the literature of younger deltaic deposits in this area, but these are limited to areas with good core/well control and so it's not clear if these deltas were from particularly significant rivers or just part of general depositional trends all along the coastal plain. Sohl and Owens (1991) suggested that Late Cretaceous deposition in South and North Carolina on the shelf represented a period of delta formation, basically spanning the Pee Dee, Cape Fear, Black and Neuse rivers. Their study focused more on the Santonian-Maastrichtian. A later paper by Prowell et al. (2003), suggests that the Cape Fear Fm. is also representative of deltaic formation for the same region. Although Prowell et al. 2003 list the Cape Fear as Turonian-Coniacian, we now know that its age is Coniacian-Santonian; however, they also interpreted deltaic deposition. The recent paper by Aleman Gonzalez et al. (2020) moves away from the deltaic model and suggests a sequence stratigraphic model for sediments of Cenomanian-Coniacian age. However, Sugarman et al. (2021) continue to espouse a deltaic model for Cenomanian/Turonian sediments in New Jersey.

*Figures 2 and 3: What is the position of the C/T boundary (dashed line) based on? Perhaps a diagonal line would be more justified? I.e., marking a range rather than a specific depth. Depths in the photographs are hard to read, a larger font would help. "Core" in the caption of Fig. 2 should be with a lower case letter.*

The C/T boundary is properly defined by the lowest occurrence of the ammonite *Watinoceras coloradoense.* In the absence of this taxon, the end of the carbon isotope excursion is a good approximation of the boundary. Thus we've placed a dashed straight line marking the boundary at the termination of the excursion (see lines 267-8 and 283-4).

*Figures 4 and 5: X-axis for the lithology is undreadable*

We've increased the font size for these.

*Figure 6: I would add "(molar)" below "C/N" and move the site names below the lowest datapoints.*

Added and moved.